# Equine Gastric Ulcer Syndrome: An Update on Current Knowledge

**DOI:** 10.3390/ani13071261

**Published:** 2023-04-05

**Authors:** Jessica Vokes, Amy Lovett, Benjamin Sykes

**Affiliations:** Equine Veterinary Clinic, School of Veterinary Sciences, Massey University, Palmerston North 4474, New Zealand

**Keywords:** Equine Squamous Gastric Disease, Equine Glandular Gastric Disease, horse, stomach, omeprazole, misoprostol, sucralfate

## Abstract

**Simple Summary:**

The term Equine Gastric Ulcer Syndrome has been used to describe mucosal diseases of the equine stomach since the 1990s. More recently, specific nomenclature has been used to differentiate diseases affecting the squamous and glandular mucosa, as the diseases of these anatomic regions vary widely. Over the past 20 years, a large amount of research has been performed to understand both diseases and their management. Significant progress has been made in their pathophysiology, treatment, and prevention. This review aims to look at previously and recently published literature to define these diseases, as well as their prevalence, diagnosis, and long-term management.

**Abstract:**

Equine Gastric Ulcer Syndrome (EGUS) is a term that has been used since 1999, initially being used to describe all gastric mucosal disease in horses. Since this time, the identification of two distinct main disease entities of the equine gastric mucosa have been described under the umbrella of EGUS; these are Equine Squamous Gastric Disease (ESGD) and Equine Glandular Gastric Disease (EGGD). In 2015 the European College of Equine Internal Medicine (ECEIM) released a consensus statement defining these disease entities. This document highlighted the lack of evidence surrounding EGGD compared to ESGD, and identified knowledge gaps for further research to be directed. Subsequently, many studies on EGGD have been published, especially on pathophysiology, diagnosis, and treatment. This article updates current knowledge on both ESGD and EGGD as understanding has evolved since the last large-scale review.

## 1. Terminology

Equine Squamous Gastric Disease (ESGD) describes lesions involving the squamous mucosa encompassing the margo plicatus, greater and lesser curvatures and the dorsal squamous fundus [1]. This can then be further classified into primary or secondary based on the known pathophysiology of disease. Primary ESGD occurs in an otherwise healthy gastrointestinal tract and is the most common form of this disease. Secondary ESGD is due to delayed gastric outflow as sequela to other diseases, such as pyloric stenosis, severe EGGD or inflammatory bowel disease (IBD) [1,2,3,4].

Equine Glandular Gastric Disease (EGGD) describes lesions of the glandular mucosa involving the cardia, ventral glandular fundus, antrum, pylorus and proximal duodenum [1]. The pylorus or antrum, often referred to as the ‘pyloric antrum’ appears to be the most frequently affected region of the glandular mucosa, as described by several gastroscopic studies [5,6,7,8].

## 2. Prevalence 

The prevalence of ESGD and EGGD has been reported extensively elsewhere [1,9,10,11]. Prevalences for ESGD vary by group sampled and exercise program. Prevalences of 37–52%, 38–56% and 48% in Thoroughbred, Standardbred and endurance racehorses, respectively, while out of training have been reported. These increase up to 100%, 72–88% and 57–93% respectively when these same populations are in training [12,13,14,15,16,17,18,19,20,21]. The prevalence ranges greatly in other populations of horses, including 11% in university teaching horses [22], 55–68% of adult horses at Thoroughbred stud-farms [23,24], 79% in Western pleasure horses [9] and 51% in Italian donkeys [25]. More recently, the prevalence of ESGD at post-mortem has been described as 61% in feral horses in the UK [26] and 64% of equids in a safari park [27]. It is worth noting that many of the prevalence studies were performed some time ago, and therefore may not be entirely relevant to current management practices.

The prevalence of EGGD is less well documented and more variable between populations. Reports include 25–65% in Thoroughbred racehorses [28,29], 47% in a mixed population of Standardbred and Thoroughbred racehorses [16], 16–33% in endurance horses [13,15], 72% in Canadian showjumpers [8], 69% in Canadian polo ponies [30], 59% in American Quarter Horses [9], 6% in Brasileiro de Hipismo military horses [31], 15% in Italian horses [32] and 3% in Italian donkeys [25]. An abattoir survey in the UK found a prevalence of 71% in domesticated horses compared to 30% in feral horses [26].

## 3. Risk Factors

Since the 2015 ECEIM consensus statement, few large-scale epidemiological studies have been performed to investigate populations at risk of EGUS. Of the studies that have been performed, many have been of limited sample size, leading to an increased risk of both type one and type two errors [33]. Therefore, results from single studies should be considered with caution and the literature looked at collectively. On this basis, several conclusions have been drawn about the risks of ESGD and EGGD for certain signalments such as breed and sex, but less so for others, such as age. The Thoroughbred and Standardbred breeds are associated with an increased risk of ESGD [34,35] although whether these are true breed predilections or simply reflective of exposure to management risk factors is unclear. Similarly, limited evidence suggests that the Warmblood breed is associated with an increased risk of EGGD; however, this is described in countries with limited breed variety [18,36] and is likely multifactorial. A small amount of evidence finds ponies at a reduced risk of EGUS when ESGD and EGGD are not differentiated [37]. There is conflicting evidence of the effect of sex on EGUS, with many studies finding geldings to be at higher risk than mares [8,9,22,30,34,38] meanwhile others finding no effect [12,13,29,38,39,40,41]. Within the adult population, age does not appear to be a significant risk factor for either ESGD or EGGD when other factors, such as exercise, are accounted for [8,18,28,30].

Investigation of ESGD and EGGD risk by management factors has been described in several publications. One recent study found a number of factors protective against undifferentiated gastric disease, including open front stabling year-round, feeding hay from a hay net (vs. unspecified), keeping retired horses as companions, and using omeprazole therapy during periods of stress [37]. A small sample of American Quarter Horses found city dwelling and use in performance disciplines to increase the risk of undifferentiated gastric disease [9]. Interestingly, this study also found the classification of horses as ‘friend’, referring to the use solely as a pet, as a risk for disease and found increasing time spent in contact with humans to be associated with higher ulcer grade [9]. 

A large-scale investigation of Thoroughbred racehorses in Australia found individual trainer, metropolitan yard location, lack of contact with other horses, solid partitions and playing talk radio to all increase the risk of ESGD [40]. Other risk factors specific for ESGD include being owned for a shorter period [30], an increased time in work, travelling [28,30,40], current training [17,19,42] or recent racing [14]. Interestingly, Standardbred trotters are at a higher risk of ESGD when directly compared to Standardbred pacers [19,42], although it is unknown if the difference in gaits, or other management factors contribute to this finding.

Crib biting is associated with an increased risk of ESGD, as are other stereotypies [28,40,43,44]. The relationship between these factors is not well studied; however, it is proposed that horses with ESGD display stereotypies subsequently to their disease [43,44]. Alternatively, it has been proposed that crib biting mimics the effects of exercise increasing the duration of acid exposure to the squamous mucosa. Thoroughbred racehorses that display aggression towards humans have been demonstrated to have a reduced risk of ESGD [28]. Factors found to be protective against ESGD in polo ponies include housing in a paddock and non-steroidal anti-inflammatory drug (NSAID) use [30].

Risk factors specifically for EGGD are less well documented. Exercising ≥ 5 days per week, racing below expectation, and trainer are risk factors for EGGD in Thoroughbred racehorses [28]. Increased days in work and actively competing are risk factors in Warmblood showjumpers [8]. Housing in single pens has also been associated with an increased risk of EGGD [37]. Although not statistically significant, an increased number of caretakers and riders showed a tendency towards an increased risk of EGGD in a mixed Finnish population [36]. The prevalence of EGGD doubles in endurance horses during the competition season, when compared to the inter-season period [13]. Collectively with the increased prevalence observed in domesticated horses, when compared with feral horses [26], this suggests that intensive management plays a role in increased EGGD risk, although via different mechanisms than for ESGD. Time in work did not affect EGGD risk in Thoroughbred racehorses [28] but increased years of experience reduces the risk of EGGD in Canadian polo ponies [30] and show jumpers [8]. This suggests an element of adaptation, management changes over time, or that the selection of healthy horses might occur.

The nutritional risk factors associated with ESGD have remained consistent since 2015. Specific risk factors include lower hay provision and consumption, lower meal number per day and higher grain and starch feeding [9,18,30,37,40,45,46]. Horses that had access to some pasture turnout were less likely to have ESGD, with horses turned out with other horses having a further decreased risk, in one study [40]. Conversely, pasture turnout was not found to be protective against ESGD in other populations [46,47]. Pasture provision likely plays a multifactorial role in disease pathophysiology that will be discussed later. Oral hypertonic electrolyte administration increases lesion number and severity of ESGD [48].

Most studies have failed to find a relationship between nutritional risk factors and EGGD [8,28,36]. A study following foals through weaning found the feeding of molasses-coated alfalfa chaff to increase the incidence of pyloric lesions when compared with hay [49]. Similarly, a recent study found alfalfa feeding to be a risk factor for EGGD [37]. The significance of these findings is unclear at this point in time and the authors caution against overinterpreting the findings pending further studies.

## 4. Clinical Signs

Many clinical signs have been ascribed to gastric mucosal disease over the years, but surprisingly few studies have investigated these on a large scale. Proposed clinical signs, discussed below, include colic [4,39,50,51,52,53,54,55,56,57], weight loss or poor body condition [4,42,51,52,56,57], poor coat condition [39], reduced appetite [4,39,51,52,56,58], diarrhea [51], bruxism [51,52], flehmen response [37], behavioral changes [43,52,56,57,59] and poor performance [14,37,39,52,56,57,60,61,62]. In interpreting the above, it is important to distinguish between ESGD and EGGD as the majority of literature is weighted towards ESGD, and a recent study found no associations between owner-reported clinical signs and EGGD [63]. Similarly, there is other evidence to show that horses with gastric disease can show no signs at all [9,14,24,28,63,64,65] and some horses with signs do not display improvement of these signs with the resolution of gastric disease [56].

### 4.1. Colic

There is evidence to suggest an increased incidence of colic and post-prandial discomfort exists in horses with both ESGD [34,39,51,53,54,57] and EGGD [34,39,51,53,54,55,57]. Supporting this, many cases show a positive response to acid suppressive therapy on follow-up, both in the literature [53,62], and anecdotally.

### 4.2. Inappetence and Weight Loss

Reports linking ‘picky’ eating, change in appetite or inappetence and EGUS exist in the literature [4,39,51,56,57,58]. Supporting this, a relationship between gastric disease and low body condition was found in Standardbred racehorses mostly diagnosed with ESGD [42] and horses with severe EGGD [4]. However, other studies failed to demonstrate a relationship between either ESGD or EGGD and poor body condition [12,28]. Interestingly, a small study in Thoroughbreds found horses with EGGD ate faster than controls [23].

It warrants note that not all cases of ESGD are primary. In cases of ESGD occurring secondary to diseases of delayed gastric emptying, such as pyloric stenosis, severe EGGD or IBD [1,2,4,28,66], it might be the primary disease process, not ESGD, that has greater influence on body condition. Association between gastritis and more distal small intestinal inflammation, suggestive of IBD [67], might explain a relationship between EGGD and weight loss.

### 4.3. Poor Coat Quality

An early cross-sectional study identified an association between a rough hair coat and gastric disease [39] but several studies have failed to demonstrate this relationship since [18,24,42]. The significance of poor coat quality or rough hair coat is questionable in its relationship with EGUS.

### 4.4. Diarrhea

In early gastric disease research, diarrhea was reported as a clinical sign in adults [51] and a more recent study found a reported increase in free fecal water to be associated with EGGD [37] supporting this finding. Other than these examples, diarrhea has not been reported as a clinical sign of gastric disease.

### 4.5. Behavior

Changes in behavior are often attributed to gastric disease in practice, although reports in the literature supporting this are variable [43,52,56,57,59]. A cross-sectional study of 50 show horses identified that horses with nervous behavior more likely to have ESGD [59]. Multiple studies have shown a relationship between ESGD and crib biting [40,43,44] or other stereotypies [28,40].

One of the common behaviors attributed to gastric disease by owners is ‘girthiness’, i.e., signs of pain and negative behavior during girthing [56,57,68,69]; however, studies have failed to show a relationship between girthing behavior and ESGD or EGGD [35,63,70].

It is hypothesized that stress plays a role in the pathophysiology of EGGD [28], and if so, this might have an association with behavior. Supporting this, horses with EGGD have increased cortisol responses to novel stimuli [23] and exogenous adrenocorticotropic hormone (ACTH) stimulation [71,72], as well as higher fecal cortisol metabolites after exposure to novel stimuli [23]. The relationship between stress and ESGD is less well-defined; one study showed an inverse correlation between hair cortisol and ESGD grade, suggesting that long-term stress is associated with ESGD [73].

These non-specific behavioral complaints are likely to have a complex, multifactorial relationship with ESGD and EGGD, where these downstream effects of pain and stress response are closely intertwined. The authors propose that gastric pain in horses could play a role in the development of stereotypical, undesired or anticipatory behaviors, and that these behaviors might continue beyond the course of disease. It is also possible that some horses are ‘predisposed’ to both gastric disease and behavioral anomalies concurrently. As there is little evidence proving causation in this area, certain conclusions cannot be made at this stage. Moreover, the behaviors attributed to ESGD and EGGD are not specific for gastric disease, thus, other gastrointestinal diseases, such as sand enteropathy [36] or IBD [74], and non-gastrointestinal (e.g., musculoskeletal) diseases should remain as differentials [36,68,69].

### 4.6. Poor Performance

Owner-reported poor performance is often reported as a clinical sign of ESGD [57], EGGD [56] or both [60,62]. Poor performance is one of the most common reported signs [56,57], despite its lack of specificity for gastric disease. Racing below expectation was a risk factor for EGGD in another study [28]. The mechanism of this effect is currently unknown, although might be due to gastric pain limiting stride-length and oxygen consumption in racehorses [75] or reducing behavioral compliance in other disciplines. Conversely, other studies failed to demonstrate this effect [14,28], possibly due to the wide scope of interpretation of performance level.

## 5. Pathophysiology

### 5.1. Equine Squamous Gastric Disease

The understanding of primary ESGD pathophysiology has not changed significantly in the years since the ECEIM consensus [1]. Management factors that increase the acid exposure of the squamous mucosa predispose to the development of ESGD [52]. The susceptibility of the squamous mucosa to hydrochloric acid and volatile fatty acids is pH, time and dose dependent [76,77]. Following initial damage by the acid, diffusion into the stratum spinosum causes ulceration [77]. By-products of bacterial fermentation of carbohydrates, such as lactic acid and volatile fatty acids, perpetuate the damage caused by hydrochloric acid, when fed in large volumes that are unlikely to be seen clinically [3,77]. It should be remembered that pasture can be a significant source of Non-Structural Carbohydrates (NSCs) [78].

The effect of pasture in the pathophysiology of ESGD is likely multifactorial, which might explain the variability of study results with some showing pasture turnout as protective [40] and others finding no effect [46,47]. Measurement of intragastric pH in horses fed hay and grain found no effect of pasture turnout in one study [79]. Specific factors in the relationship between ESGD and pasture are not defined in equids at this time. Likely factors include the NSC content, fiber quantity and fiber composition of the pasture provided. The importance of fiber in the pathogenesis of ESGD is likely two-fold; by increasing the saliva produced by chewing, which has a buffering effect on stomach acid, and its ability to create a ‘roughage ball’ in the stomach to limit acid splashing. Therefore, it follows that not all fiber will have the same effect, with fiber size likely being an important factor, as it is in creating a ruminal mat and increasing time masticating in cattle [80]. Other effects of diet on the microbiota of the stomach, and its relationship to disease are not yet fully understood. It is also possible that other factors associated with pasture turnout are associated with risk reduction, such as increased socialization [40], display of natural behaviors and stress reduction.

Exercise increases the exposure of the squamous mucosa to acid due to increased abdominal pressure and stomach contraction [81]. In fitting with this, prevalence and ESGD score is associated with the intensity of long-duration exercise in Thoroughbreds [19,82,83] and distance of ride in endurance horses [13].

Secondary ESGD can occur due to delayed gastric outflow from other diseases, including pyloric stenosis and severe EGGD [1,2,3,4]. The possibility of severe EGGD causing delay in gastric emptying, and subsequently ESGD, has been proposed by several authors in recent years [2,4,28]. Diagnosis of delayed gastric emptying is difficult in horses [2], which makes confirmation of this hypothesis scant despite its theoretical validity [84].

### 5.2. Equine Glandular Gastric Disease

Conversely, the pathophysiology of EGGD remains poorly understood. It has been hypothesized that the damage to the glandular mucosa in cases of EGGD is due to a loss of normal defense mechanisms to physiologic acid [1]. In humans, this is most often caused by infection with *Helicobacter pylori* and NSAID use [85]. In horses, there is evidence to suggest that EGGD is a form of gastritis, identified by histopathology [67,86,87,88]. One study even showed a correlation between histopathological gastric glandular inflammation to duodenal inflammation, including lymphoplasmacytic inflammation and eosinophilic infiltrate, but not to more distal segments of the GI tract [67]. This evidence of EGGD as a form of gastritis is supported by evidence of immune upregulation in EGGD, demonstrated by altered protein composition of serum and saliva of horses with EGGD compared to controls [89]. Further work is needed to deepen the understanding of inflammation in EGGD pathophysiology.

### 5.3. Non-Steroidal Anti-inflammatory Drugs

Research into NSAIDs as a common cause of ESGD or EGGD at the population level is lacking. Furthermore, several population-based studies have failed to find NSAID use associated with increased risk of EGGD [28,30,90]. At an experimental level, multiple studies have shown the ulcerogenic capacity of NSAIDs at high doses in horses [50,91,92,93,94,95,96,97]. Most of these studies, however, use NSAID doses in excess of what is typically recommended for clinical use [98]. The ability for fed-fasted NSAID models to cause both ESGD and EGGD at normal therapeutic doses has also been demonstrated [92,97]. These models are worth considering when assessing the risk of EGUS in hospitalized horses that often undergo periods of fasting with concurrent NSAID administration. A study investigating phenylbutazone given at 4.4 mg/kg daily for 10 days showed an increased risk of EGGD compared to firocoxib at 0.1 mg/kg daily, which was itself increased compared to the control [99]. Conversely, another study looking at label dosing of phenylbutazone and suxibuzone for up to 15 days failed to cause increased risk of EGGD [100]. The mechanism of NSAID-associated EGGD is not fully understood at this time, with prostaglandin concentrations in the stomach and glandular mucosa not changing when NSAID-associated disease was induced in a single study [91]. The authors believe that the ability for NSAIDs to cause EGGD is over-estimated in the clinical setting and that the use of high-dose NSAID models for induction of EGGD for treatment trials is not justified; instead, these trials should focus on naturally occurring disease.

### 5.4. Helicobacter spp.

*Helicobacter* species have been implicated in gastric and duodenal ulceration in humans since the 1980s [101]. *Helicobacter*-like species have been identified in the stomachs of horses with EGUS; however, studies fail to associate these bacteria with gastric disease [102,103,104,105,106,107]. Other studies have failed to detect *Helicobacter*-like species from horses with gastric disease [108,109,110,111]. As such, the authors believe that at present there is no evidence to support that *Helicobacter* species play a role in EGGD pathogenesis.

### 5.5. Microbiota

Recent research has shown that horses with unclassified gastric disease have lower gastric and fecal microbial diversity compared to healthy controls [112]. One study looking at microbial communities from gastric fluid and mucosal biopsies found differences in the microbiome between horses with gross EGGD and controls [102]. Another study comparing diseased and non-diseased areas of glandular mucosa within horses showed significant differences in *Firmicutes* and *Proteobacteria* [111]. Stomach bacterial diversity was shown to cluster by bedding type, water access and feeding frequency in one study [108], and by management involving stabling compared to pasture turnout in another [109]. A recent study showed glandular mucosal microbiota to differ with multiple management factors including offering hay, type of hay, provisions of ‘sweet feed’, turnout and stalling [113]. In humans, an increase in the *Firmicutes* phylum, specifically the *Streptococcus* genus, is associated with non-*H. pylori*, non-NSAID associated gastritis [85]. Increasingly, the relationship between microbiota and EGGD pathophysiology appears to be relevant, although unclear, and the definition of a ‘healthy’ microbiota is not established at this time. Importantly, although microbial differences exist between diseased and non-diseased animals, the authors do not believe that this supports the use of antimicrobials for the treatment of EGGD at this point in time. Consistent with this, one study showed no improvement in gastric disease with an oral antimicrobial in combination with omeprazole [114].

### 5.6. Management

Increased days of work per week is a risk factor for EGGD in both Thoroughbred racehorse and Warmblood showjumper populations [8,28]. The prevalence of EGGD doubles in endurance horses during the competition season [13]. These findings suggest that exercise plays a role in the pathophysiology of disease. A proposed hypothesis for this is the disruption of normal blood flow to the stomach during exercise, with exercise acting as a physiological stressor for the glandular mucosa [115].

Several studies have found an association between trainers and EGGD risk [28,36]. The reason for this is unclear.

One proposed mechanism is through environmental stressors with the domestication and management of horses potentially playing a role in the pathophysiology of disease [28]. The findings of pet horses being at increased risk of disease, as well as increased time spent with humans associated with increased ulcer grade in one study [9] support this theory and fit with the higher prevalence of EGGD in domesticated horses compared to a feral population seen in another [26]. Horses diagnosed with EGGD have higher cortisol responses to novel stimuli [23] and exogenous ACTH [71], as well as higher fecal cortisol metabolites after exposure to novel stimuli [23], further supporting a role for behavioral stress in the pathogenesis of disease.

## 6. Diagnosis

Gastroscopy is consistent and reliable in the diagnosis of squamous disease [116]. However, there are growing concerns regarding the significance of gross glandular lesions to their presenting complaints [63] and histological findings [88]. Gastroscopy has been shown to have moderate inter- and intra-individual agreement for ESGD [116], but considerable variability for descriptors of EGGD [116,117,118]. Further, there is also increasing evidence of the lack of association between gross glandular lesions and histological evidence of inflammation [67,88]. Therefore, the use of gastroscopy alone to determine clinical significance of lesions should be avoided, especially with regards to EGGD. Instead, other factors, such as owner or trainer reported complaints and clinical signs and response to therapy, should be considered alongside gastroscopy to assess the potential relevance of gross mucosal changes.

The use of histopathology to assess disease, especially of the glandular mucosa, is becoming increasingly described [67,88,119,120]. One study directly comparing gross EGGD to histopathological disease showed poor correlation, with 71% of grossly normal stomachs having mild gastritis, and all EGGD lesions demonstrating various degrees of gastritis histologically, regardless of gross severity [88]. Another study demonstrated both glandular gastric lymphoplasmacytic inflammation and eosinophilic infiltrate in a relatively small sample population [67]. The comparison of biopsy techniques feasible via endoscopy to full thickness samples post-mortem found the ‘double bite’ technique to yield the best samples for assessment [88]. Larger samples for histopathology are described using a snare when lesions are sufficiently raised, such as glandular polyps (Figure 1) [121].

Alternative means of diagnosis aside from gastroscopy are appealing. The use of fecal occult blood testing for ESGD and EGGD was first described over a decade and a half ago, initially showing average performance in diagnostic ability [122]. Further independent studies have been performed, indicating that fecal occult blood testing is unreliable for the diagnosis of EGUS [123,124,125]. Similarly, studies show that hematology and biochemistry or inflammatory markers, such as serum amyloid A (SAA), have little clinical use in the diagnosis of, or screening for either ESGD or EGGD [126,127]. Research into novel serum and salivary testing for detecting ESGD and EGGD has been undertaken with some differences demonstrated between diseased and control populations, although a large degree of overlap exists between populations [50,89,126,128,129]. The clinical utility of these tests is very limited at this point in time.

Initial investigation into sucrose permeability testing in a feed deprivation model of disease found a correlation of urinary sucrose with ESGD with 83% sensitivity and 90% specificity [130]. Further research into sucrose permeability testing by serial blood sampling initially showed a difference between moderate or marked ESGD compared to the baseline [131]. The same author then applied sucrose permeability testing to weanling foals with gastric disease, finding a sensitivity of 81–97% with a poor specificity [132]. In contrast, the same protocol in a wider sample of adults showed poor sensitivity and specificity for the detection of ESGD or EGGD compared to healthy controls [6]. Another showed no association of sucrose permeability testing with ESGD or EGGD [133].

## 7. Grading

The continued use of the 0–4 scale described by the EGUS council [134] for the description of ESGD remains an effective way to classify and monitor disease and allows easy assessment of large populations and comparison between studies. Anecdotally, the authors note that some horses with grade 1/4 ESGD and clinical signs respond to therapy, while other horses with grade 4/4 disease show no change in clinical signs with treatment. Therefore, the usefulness of a numerical scale can be limited when applied at the individual patient level and the authors propose that it might be more appropriate to consider disease on a dichotomous present or absent basis, consistent with the ECEIM consensus statement recommendations [1].

Grading lesions of the squamous mucosa as described by Sykes et al. [1], adapted from Andrews et al. [134]:Grade 0 Epithelium intact, no appearance of hyperkeratosisGrade 1 Mucosa intact, areas of hyperkeratosisGrade 2 Small, single or multifocal lesions (Figure 2)Grade 3 Large single or extensive superficial lesionsGrade 4 Extensive lesions with areas of deep lesions

For EGGD, the recommendation at this time remains not to assign a grade to these lesions, but instead, that lesions be described by anatomical location, distribution, severity and appearance [1]. Appearance is described as hyperemic/hemorrhagic (Figure 2 and Figure 3), erosive/ulcerated or fibrinosuppurative, and by contour; depressed, flat or raised [1]. The existence of glandular gastric polyps is also considered as a form of EGGD [120,121].

## 8. Treatment

The pathophysiology of ESGD is sufficiently well understood and studied to support acid-suppression therapy as the basis of pharmaceutical therapy [1]. Conversely, the treatment of EGGD is not as well understood. Without fully understanding the pathophysiology of EGGD, it is difficult to create treatment recommendations based on the primary cause. There is a body of evidence showing that acid-suppression therapy, in combination with mucosal protection, is efficacious in the treatment of EGGD, albeit with less success when compared to the treatment of ESGD [29,123,135,136].

Healing rates for oral omeprazole monotherapy range from 67–100% for ESGD [135,137] and up to 100% using long-acting injectable omeprazole [5,138] over a 2–4 week period. Treatment for ESGD remains as previously recommended [1] with proton pump inhibitors (PPIs), such as omeprazole, as the cornerstone of therapy and other agents, such as H2 pump blockers, used if resolution is not seen.

In contrast to ESGD, oral omeprazole monotherapy has lower rates of EGGD healing, at 14–25% [29,135,138]. However, up to 93% EGGD healing has been reported for a long-acting injectable formulation [5,57,66]. It has long been recognized that, no matter the cause of EGGD, reducing the acidity of the stomach can allow for healing of the glandular mucosa [1]. The efficacy of treatment for EGGD with oral omeprazole is improved by adjunctive mucosal protection such as oral sucralfate [70].

### 8.1. Omeprazole

The main changes to acid suppression therapy in recent years is the licensing of more omeprazole formulations for horses. Different formulations of omeprazole paste have differing pharmacokinetic profiles, and thus, different dosing recommendations [1]. Literature on the relative pharmacokinetics of different formulations of oral omeprazole in horses is growing. A study comparing the pharmacokinetics between five oral omeprazole concentrations found a buffered formulation and two enteric coated granule formulations to have no differences to the reference enteric coated granule formulation [138]. Another study directly compared the pharmacokinetics and pharmacodynamics of an enteric coated granule-in-paste formulation with a buffered formulation at 4mg/kg and found no differences [139]. In contrast, an enteric coated formulation was found to have a 26% higher bioavailability in a further study [140].

The importance of minor variations in bioavailability are unclear in the clinical setting but care should be taken in extrapolating the results of one formulation to another. Instead, the authors propose that clinical endpoints are likely a better marker for comparing potential differences between formulations. A clinical study in Thoroughbred racehorses found enteric coated omeprazole granules-in-paste formulation at 1, 2 and 4 mg/kg per os (PO—orally) semel in die (SID—once a day) to not differ in their improvement and healing rates for both ESGD and EGGD [29]. This is in contrast to an earlier study that found dosing of a buffered formulation of omeprazole at 4 mg/kg PO SID to be superior to a 1.6 mg/kg PO SID dose in ESGD, but not EGGD, healing [135]. These findings support the use of formulations at their registered dose, regardless of formulation type, as the registration process typically accounts for the interaction between variations in dose and bioavailability between formulations.

Oral omeprazole has superior bioavailability when fed to fasted animals [141]. It is also a pro-drug that is absorbed into systemic circulation and requires activation in response to feeding before reversibly binding and inhibiting proton pumps in the stomach [142]. As such, it is preferred that omeprazole is given after a period of fasting and prior to a meal [29,115,135,143]. Due to the normal circadian rhythm of feed intake in horses, removing feed to enforce an overnight fast has minimal effect on gastric pH [144]. Therefore, it is recommended to give omeprazole early in the morning after a period of overnight fasting, a minimum of 30 min before the re-introduction of feed (see Figure 4 and Figure 5) [20,115].

Given the role of exercise-associated gastric contraction and acid splash in the pathophysiology of ESGD [81], as well as the duration of acid suppression not spanning a full 24 h [143,145], it is logical that timing the omeprazole treatment pre-exercise would be superior for the healing of ESGD lesions [29]. A single study comparing treatment timing found a trend for the timing of feeding to affect the outcome; however, this was not statistically significant, likely due to an inadequate population size [135]. Considering this, the effect of omeprazole timing should not be dismissed at this time and the authors recommend administering omeprazole followed by a small feed prior to exercise to optimize acid suppression at the time of peak injury to the squamous mucosa.

Earlier investigations into long-acting intramuscular omeprazole at 7-day intervals for 2 weeks found the resolution of 52–100% and 54–75% of ESGD and EGGD, respectively [5,146]. A further 2 weeks of treatment increased resolution rates to 86% and 58% of ESGD and EGGD, respectively [146]. Two small parallel retrospective studies comparing 4 mg/kg IM long-acting injectable omeprazole weekly to conventional oral omeprazole found the injectable formulation to be non-inferior for the treatment of ESGD and EGGD [57]. A recent study showed superior results using a compounded, long-acting intramuscular formulation (currently available in Australia, the UK and Canada) at 5-day intervals compared to 7-day intervals, with 97% and 93% resolution of ESGD and EGGD, respectively, over 4 treatments [66]. In contrast, a study comparing the compounded formulation of long-acting injectable omeprazole available in the Unites States, given weekly, found no difference to conventional oral omeprazole treatment [147]. This formulation is also responsible for a higher rate of injection site reactions at 8% after the first injection, increasing to 48% at the 4th injection [147], compared to 1.2–6.5% over the treatment course for the Australian/UK/Canadian formulation [56,57,66,146].

The duration of acid-suppressive therapy should also be considered. Evidence supports that 3 weeks of treatment is sufficient for ESGD treatment [137] and that if healing has not occurred in that time, then the benefit of prolonged treatment is unclear. Similarly, recent reports suggest that 3–4 weeks of treatment might be sufficient for EGGD if adequate acid suppression can be achieved [66].

### 8.2. Sucralfate

Sucralfate is a complex polyammonium hydroxide salt that adheres to the glandular mucosa. Proposed benefits of this treatment, additional to acting as a physical barrier to acid diffusion, includes the stimulation of mucus secretion, inhibition of pepsin and bile-acid release, prevention of fibroblast degradation, stimulation of growth factors and increased production of prostaglandin E [115]. With the uncertain and likely complicated pathophysiology of EGGD, the addition of a mucosal protectant to acid suppression therapy for this disease is logical, consistent with the current Consensus Statement recommendations [1]. Supporting this, a combined omeprazole and sucralfate therapy for the treatment of EGGD is superior to omeprazole alone [148]. The recommended dose is currently 12 mg/kg PO bis in die (BID—twice a day) [1], although many authors report doses up to 20 or 30 mg/kg three to four times per day [97,149]. Comparisons of the clinical efficacy of different doses and frequency on naturally occurring disease have not been reported.8.3. Alternatives to Omeprazole

Limited alternatives to omeprazole therapy have been investigated, with sparse published efficacy data available. The three best studied alternatives include a prostaglandin E analogue (misoprostol), another PPI (esomeprazole) and drugs belonging to the H2 receptor antagonist class, such as ranitidine.

### 8.3. Esomeprazole

In recent years, the proton pump inhibitor esomeprazole has been further investigated both in models and naturally occurring disease. A pharmacodynamic study comparing two doses of esomeprazole and two diets found that doses of 2 mg/kg PO SID when fed ad lib hay or 0.5 mg/kg PO SID when fed a high-grain, low-fiber diet sufficiently increased gastric pH to above the threshold for mucosal healing [150]. Two small clinical reports exist describing esomeprazole use for the treatment of EGGD in clinical practice [151,152]. The first found a healing rate of 80% in five horses previously refractory to omeprazole treatment, when given at 0.5 mg/kg PO SID [151], and the second found 67% healing in three horses treated for 14 days and 75% in four different horses treated for 28 days, when given at 2 mg/kg PO SID [152]. These findings suggest the potential application of esomeprazole, at a suitable dose, as a second line treatment if failure occurs with first line therapy; however, its efficacy in direct comparison to omeprazole in clinical cases is not reported. Another small study showed doses of 40 mg and 80 mg per horse to increase gastric fluid pH above 4 over a 6–hour period [153]. Although a brief clinical response might be seen at these doses, they are not recommended for treatment based on a more comprehensive study [150].

### 8.4. H2 Receptor Antagonists

The H2 receptor antagonists ranitidine and famotidine have been shown to suppress stomach acid in horses experimentally [154]. However, this effect is variable between horses, doses and in its duration of acid suppression [154]. Studies show ranitidine to reduce ESGD prevalence in racehorses over 4 weeks; however, this effect is inferior to omeprazole [21,155] and EGGD scores did not differ [21,155]. Furthermore, ranitidine is not commercially available in many places, and the published efficacy for other H2 receptor antagonists is lacking. Thus, omeprazole remains the drug of choice for acid suppression in horses.

### 8.5. Misoprostol

The mechanism of action of misoprostol in EGGD is not fully understood [10]. Prostaglandin E is thought to have various roles in the normal protection of the glandular mucosa as demonstrated in other species, including enhancing mucosal blood flow, increasing bicarbonate and mucus secretion and reducing acid production [17,156]. It has also been shown to have anti-inflammatory effects on equine leukocytes [157,158] in vitro. Misoprostol is protective against NSAID toxicity and is currently a treatment of choice for NSAID induced colitis [159,160]. There is limited evidence of the efficacy for prostaglandin analogues in the treatment of EGGD, especially in comparison to the more common omeprazole and sucralfate regimens. A single non-randomized study found misoprostol to be superior to omeprazole-sucralfate therapy for the treatment of EGGD in a modestly sized population without strict controls such as dosing omeprazole to non-fasted animals [161]. The authors consider that misoprostol is a promising treatment option for EGGD; however, more evidence needs to be gathered before considering it as a first line therapy. It warrants note that it can cause adverse effects to humans including fetal loss [162] and that care should be taken in its handling by both veterinarians and clients.

### 8.6. Risks of Omeprazole

The use of omeprazole in humans is associated with complications, including rebound gastric hyperacidity [163,164,165], increase in antimicrobial-associated and non-specific diarrhea risk [166] and increased fracture risk in humans [166,167,168]. The mechanism described for rebound gastric hyperacidity in humans is through increases in serum gastrin during PPI treatment. This effect has been demonstrated in horses with as little as a 7-day treatment [169]. Similarly to humans, an increased risk of non-specific diarrhea is seen in foals treated with omeprazole [170].

A recent study looking into the concurrent administration of phenylbutazone and omeprazole found a significant increased risk of GI complications, affecting 75% of horses in the concurrent treatment group [96]. These complications included a death and a euthanasia due to complications of entero- and typhlo-colitis, a case of undiagnosed colic, two impactions and a case of diarrhea [96]. Alterations in intestinal motility, exacerbation of NSAID-associated dysbiosis, inflammation or ulceration have been proposed as mechanisms [171]. This is consistent with findings in dogs showing increased intestinal inflammation when omeprazole is given concurrently with carprofen [172]. Conversely, limited studies performed to date show no effect on the fecal microbiota following 7 and 28 days of omeprazole treatment [173,174]. Collectively, these findings highlight the importance of ongoing research into the concurrent use of NSAIDs and PPIs in horses, a common practice.

Concerns have also been raised about the potential for omeprazole to increase fracture risk in horses [175] and omeprazole has been demonstrated to reduce calcium absorption in horses [176]. In one study, 8 weeks of omeprazole treatment at 1 mg/kg PO SID in a group of horses showed no difference in fecal and urine mineral balance, radiographic bone aluminum equivalence, markers of bone formation or other skeletal health markers compared to healthy controls [177]. This study was performed on adult horses that were not undergoing forced exercise programs, so extrapolation to the young racing population cannot be made at this stage.

The use and risks of omeprazole use have recently reviewed elsewhere [175]. Further research is needed to further understand the potential complications surrounding both short- and long-term omeprazole use in horses and consideration should be made before making treatment recommendations.

## 9. Discontinuation of Acid Suppressive Therapy

One concern with omeprazole therapy is the recurrence of disease following the discontinuation of treatment, and a recent study has demonstrated that ESGD prevalence can return to pre-treatment levels in as little as 3 days following discontinuation of treatment [20]. Rebound Gastric Hyperacidity (RGH) has been proposed to play a key role in the rapid recurrence of disease in human medicine [178]. The mechanism for RGH by the increase in intragastric pH causes a loss of negative feedback on D-cells, causing hypergastrinemia [178]. Gastrin acts on the Enterochromaffin-like (ECL) cells of the stomach, causing histamine release, as well as having a trophic effect on the ECL cell population [179]. It is hypothesized that both increased ECL cell density and hypergastrinemia contribute to increased acid secretion, termed RGH [180]. Previously, it has been shown that serum gastrin doubles over a 14-day course of omeprazole [176], further supporting this hypothesis in horses.

Recent work by the authors [169] showed that omeprazole causes a >2-fold increase in serum gastrin concentrations within 7 days of treatment. No further increase was seen over an 8-week treatment period and gastrin concentrations returned to baseline within days of discontinuation. The same study also looked at serum Chromogranin A (CgA) as a proxy for ECL cell populations, as is used in human medicine [181]. Omeprazole treatment and discontinuation did not influence CgA concentration [169]. Collectively, the findings of the recent work suggest that a brief period of RGH might occur within a 48–hour window following the administration of the last dose of omeprazole, but that the effect is not prolonged. As such, the authors currently do not recommend tapering of omeprazole for treatment durations of ≤8-weeks. Instead, the authors focus on the 24–48-h period following the administration of the last dose of omeprazole, making sure that horses are provided with appropriate roughage during this time, and that they are not exercised nor transported during the expected, albeit brief, RGH event that might occur in this window. Further studies are needed to evaluate the impact of longer duration treatment on serum gastrin and CgA concentrations with tapering recommendations adjusted accordingly.

## 10. Prevention

Prevention of both ESGD and EGGD requires the management of multiple factors. These can be split into broad categories of management, supplementation with nutraceuticals and pharmaceutical use. Management changes can have the largest impact on prevention of both ESGD and EGGD, and as such should be the foundation of any preventative strategy, with nutraceuticals, then pharmaceuticals reached for sequentially. It is also noted that few recommendations for the prevention of ESGD and EGGD have been well studied in real-world populations with most derived from proposed pathophysiology and risk factors.

### 10.1. Management

The management factors that have the most impact in the prevention of ESGD are roughage, exercise and the NSC content of the diet. Providing ad libitum roughage has been a longstanding recommendation for the prevention of ESGD [1]; however, horses have been shown to have a circadian rhythm of foraging [144,182,183] and pasture turnout has been shown to have inconsistent effects on ESGD [30,40,47]. These variables are likely to limit the effectiveness of simply providing ad libitum forage, especially if pasture based, and more nuanced strategies are required for optimal efficacy. The authors recommend that at least 2% BWT/per day of good quality roughage should be consumed. Importantly, the simple provision of ad libitum roughage does not ensure adequate intake, and one simple, early step in investigating unexplained ESGD is to measure the actual roughage intake to ensure that this threshold is being met.

Both timing and duration of exercise play an important role in ESGD risk. The timing of exercise should be considered when implementing a preventative strategy for ESGD and it is logical that horses should be exercised when there is maximal intra-gastric buffering. Due to the tendency of horses to consume most of their roughage intake during daylight hours [144], the most logical time to exercise is in the afternoon when normal protective mechanisms will be at their peak. Other strategies to reduce ESGD risk include the use of multiple hay nets [182] or feeding highly palatable hay prior to exercise, with as little as 300g appearing to have a significant buffering effect [144]. Limiting the cumulative duration of exercise at or above a trot to an average of less than 40 min per day [30] is also recommended for ESGD prevention. In contrast, duration of exercise is not a risk factor for EGGD. Instead, the number of exercising days per week has been demonstrated as a risk factor [8,28]. To address this the authors currently recommend ensuring that horses predisposed to EGGD get at least two, and ideally three full rest days per week.

Although high NSC diets are a well-documented risk factor for ESGD, several studies suggest a limited effect of low NSC diets in preventing ESGD in real-world environments [20,46,184]. This suggests that the impact of NSC on ESGD risk is relatively low compared to the impact of roughage intake and exercise. Regardless, minimalization of NSC intake is considered a principle of good equine nutrition [1], supported by one study that demonstrated a decrease in both ESGD and EGGD in horses with dietary starch reduction compared to controls [185].

There is little evidence to support specific management recommendations to prevent EGGD. Therefore, recommendations are often extrapolated from, and aimed at reducing risk factors associated with EGGD. In addition to ensuring adequate rest days, it is recommended to reduce the number of handlers/riders and ensure cohabitation with other horses. With the likely role of behavioral stress in the pathophysiology of EGGD it is logical to recommend reduction of stressors wherever possible [115] (e.g., by providing a competition-free, consistent and enriched environment, and through allowing horses to display natural behaviors (i.e., grooming)).

Despite recommendations surrounding the prevention of ESGD and EGGD being made for several decades, studies looking at their effectiveness are widely lacking. Further work is needed in this area to identify effective preventative strategies.

### 10.2. Nutraceuticals

Increasing interest in nutraceuticals to both heal and prevent EGUS has been seen in recent years. The addition of vegetable oil to the diet of horses has been a practice for over 30 years [186] and more recently has gained interest in relation to EGUS, and more specifically EGGD [1]. Corn oil has been shown to increase the prostaglandin and n-acetylglucosaminidase concentrations and pH of gastric fluid [149,187]. The increase in prostaglandin and n-acetylglucosaminidase observed have an unknown effect on the stomach and might be considered contraindicated in inflammatory conditions such as gastritis. In an NSAID model of disease, corn oil was found to heal EGGD similarly to sucralfate, with no effect in ESGD [149]. Similarly, a small cross-over trial in healthy mares found no effect of three vegetable oils on ESGD in a feed deprivation model of disease [188]. The use of vegetable oil in naturally occurring EGUS has not been investigated on a large scale, but it is a logical way to reduce dietary NSC content, with doses up to 1 mL/kg per day well tolerated.

Due to its wide availability in the Americas, studies looking at corn oil supplementation have dominated the literature; however, recently the feeding of omega-3 rich sources of oil have become increasingly popular. This is logical, especially for EGGD, as it has been well established in human medicine that the dietary poly-unsaturated fatty acid (PUFA) ratio contributes to the regulation of systemic inflammation [189,190]. Therefore, the type of oil used for dietary supplementation warrants consideration.

Fatty acids are considered long-chained (LC-PUFA) if they contain a carbon skeleton of 20 atoms or more; short-chain PUFAs (SC-PUFAs) contain a skeleton of 18 carbon atoms or less. Within the classification of SC- and LC- PUFAs, there is a further classification by the location of the first carbon skeleton double bond, as n–3, n–6 or n–9, associated with the third, sixth or ninth carbon, respectively. Examples of n–3 LC-PUFAs include eicosapentaenoic acid (EPA), docosapentanoic acid (DPA) and docosahexaenoic acid (DHA). These are found commonly in oily fish as well as novel algae-based supplements. An example of a n–3 SC-PUFAs is alpha-linolenic acid (ALA), found in flax. N–6 SC-PUFA examples include linoleic acid (LA) and gamma-linolenic acid (GLA), found commonly in corn and soy. Oleic acid (OA) is an n-9 monounsaturated fatty acid (MUFA) that is found commonly in canola and olive oil. It should be cautioned that it is the ratio of these oils in the total diet that contributes to their inflammatory modulation, not their absolute values in any given feed.

In human medicine, it has been shown that n–3 PUFAs decrease the inflammatory response when compared to n–6 PUFAs in several diseases [190], including IBD [189]. Specifically, an increased ratio of n–3 LC-PUFAs has been shown to have the most potent anti-inflammatory benefit [190]. There is limited conversion from dietary n–3 SC-PUFAs to n–3 LC-PUFAs [191,192], and this is where the emphasis on increasing the intake of n–3 LC-PUFAs through fish oil supplementation has gained popularity. A recent study evaluating the supplementation of exercising Thoroughbreds showed n–3 LC-PUFAs derived from fish to be superior at protecting against ESGD and altering inflammatory profiles, when compared to combined n–3 and n–6 SC-PUFAs derived from a corn and flax blend [193]. Further work looking into the inflammatory benefits of different, common dietary fats in horses, and their application in EGGD would be beneficial. However, it is logical to apply principles from other species and make recommendations to prioritize the supplementation of n–3 LC-PUFAs, such as in highly palatable fish-oil based supplementation, when possible, then n–3 SC-PUFAs secondarily, such as in flax-based products, with n–6 and n–9 PUFA based oils reached for lastly.

Pectin-lecithin complexes have shown promise in healing for both ESGD and EGGD lesions. Many small trials using pectin-lecithin-based in-feed supplements show improvement in ESGD [194] and both EGGD and ESGD [195]. Feeding of beet pulp is a source of dietary pectin and its inclusion in the diet has been shown to reduce the risk of ESGD [8]. The addition of buffering to coating agents has been proposed to be synergistic and a randomized, placebo controlled trial in horses found a dietary supplement containing a pectin-lecithin complex, *Saccharomyces cerevisiae* and magnesium hydroxide to be protective against ESGD and EGGD, following treatment with omeprazole [196]. These findings are supported by a study investigating a similar formulation wherein the combination reduced the recurrence of ESGD during Thoroughbred racing withholding periods [197]. A small trial found that *Saccharomyces cerevisiae* as a sole ingredient reduced anaerobic bacteria and amylolytic concentrations at two doses, as well as decreasing lactate-utilizing bacteria at higher doses in horses fed a high-starch diet [198]. This suggests that *Saccharomyces cerevisiae* may have utility in reducing the negative impact of a high-starch diet on the stomach as well as its benefits to the hindgut [199,200], but its clinical application is yet to be fully investigated [198] and that it is likely best utilized in combination with other ingredients.

Licorice is a popular ingredient in many nutraceuticals with proposed benefits, and a recently published study showed that feeding *Glycyrrhiza glabra* (licorice) root extract at 17.6 mg/kg once daily significantly reduced the severity of phenylbutazone-induced EGGD in miniature donkeys [201]. Daily supplementation of a mix including licorice root extract, magnesium hydroxide, calcium carbonate and aloe vera was shown to significantly reduce the number and severity of squamous and glandular lesions in a small population of horses in high level training without controls [202]. The testing of a product containing alfalfa, calcium carbonate, magnesium hydroxide, aloe vera and licorice extract on seven racehorses found a reduction in ESGD scores over 30 days; however, this was not compared to a control group [202].

Several other nutraceuticals have been studied in small, single studies. A micronized, fermented and pasteurized soy product was evaluated in a population made up of a majority of racehorses, and was protective against increased ESGD scores compared to the pollard control fed for 30 days [203]. A non-blinded study found that one month’s supplementation with a fermented rice extract significantly improved naturally occurring ESGD scores compared to tap water [204]. Aloe vera fed at 17.6 mg/kg PO BID for 28 days was found to reduce naturally occurring ESGD scores and heal lesions in 56% of 19 horses in one study without a control group [136]. A feed deprivation model found a supplement containing sea buckthorn berries to protect against worsening EGGD seen in the control group, but not against ESGD [205]. A small study looking at supplementation with a novel hyaluronan and schizophyllan showed a significant improvement of both ESGD and EGGD pre-existing lesions, without controls [206]. A randomized, blinded trial found a patented, alkaline mixture of salts to heal naturally occurring ESGD in trotters [207]. A small trial investigating an exercise induced model of ESGD found serum-based bioactive proteins from bovine serum to be protective against lesion development compared to controls [208]. A combination of turmeric and devil’s claw did not significantly increase healing of naturally occurring non-specified EGUS [209]. A mixture of Chinese herbs was not found to significantly reduce ESGD scores or increase gastric pH in a fasting model of disease [210].

Many of the studies evaluating nutraceuticals for ESGD do not account for variabilities in feeding and management changes in different populations. Further, many studies use non-clinical models of disease, the relevance of which to clinical disease, which is multi-factorial, is questionable. Despite the positive effects shown by some nutraceuticals, the importance of diet in the management of gastric disease should not be undervalued and the authors’ preference is to use nutraceuticals as an adjunct to management changes or where management changes are not effective. Preference regarding the selection of nutraceuticals should be for formulations with published studies supporting efficacy in naturally occurring disease, including appropriate controls.

### 10.3. Pharmaceuticals

The use of pharmaceuticals for the prevention of ESGD and EGGD should be the final consideration following the implementation of management changes and use of nutraceuticals, where appropriate. The need for the different levels of preventative intervention varies between population and horses, depending on a wide range of factors described above that affect susceptibility to disease.

The use of omeprazole in the prevention of ESGD is a well described and common practice [211]. The idea of a ‘set and forget’ preventative plan using omeprazole should be avoided whenever possible, as this strategy has been shown to fail in more than 20% of Thoroughbred racehorses [211]. Further, omeprazole has been shown to induce its own metabolism in humans and, consistent with this, omeprazole has shown to have a reduced area under the curve (AUC) and maximal concentration (Cmax) on day 29 of treatment when compared to day 1 in horses [212]. Another study showed a decreased efficacy for ESGD prevention in horses treated for 90 days [213]. Therefore, the long-term use of omeprazole to prevent ESGD is not recommended without the significant application of other management factor alterations first and appropriate monitoring.

The use of long-term, low-dose omeprazole therapy for the prevention of EGGD has not been validated. Furthermore, pharmacodynamic work suggests that low-dose omeprazole is unlikely to consistently induce adequate acid suppression to have protective effects at the level of the glandular mucosa [143]. Instead, the authors’ preference is the use of ‘pulse therapy’ with PPIs only administered at times of significant risk, such as competition or long-distance travel. This approach is supported by a recent study that demonstrated that using omeprazole therapy during periods of stress decreased the prevalence of undifferentiated EGUS [37].

## 11. Conclusions

This review highlights the lack of, and sometimes conflicting, information surrounding gastric mucosal disease in horses. The risk factors, pathophysiology, diagnosis, treatment and prevention appear to be relatively well understood and agreed upon for ESGD, although reported clinical signs appear to be varied. Conversely, information regarding the pathogenesis, risk factors, diagnosis, treatment and prevention of EGGD is lacking.

Due to the unknowns surrounding gastric disease in horses, many recommendations are made by the extrapolation of fact and are therefore subject to change as more is learned. Future research should be aimed at the relationship between behavior, stress and EGGD, as well as looking at preventative strategies, particularly at times of increased risk such as prolonged NSAID use.

## Figures and Tables

**Figure 1 animals-13-01261-f001:**
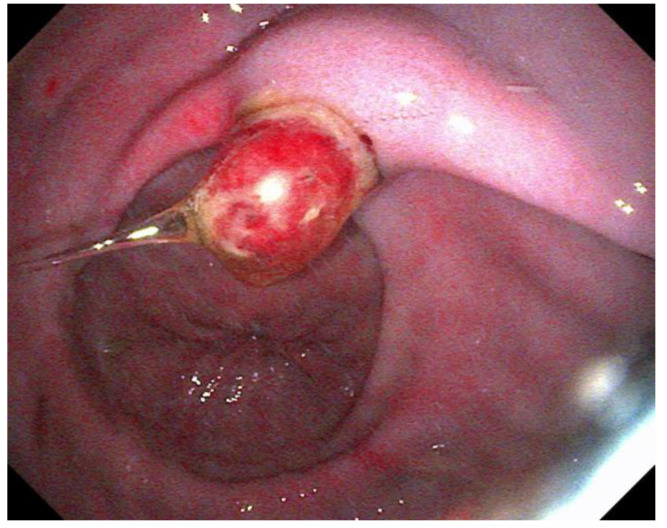
An adenomatous polyp in the pyloric region.

**Figure 2 animals-13-01261-f002:**
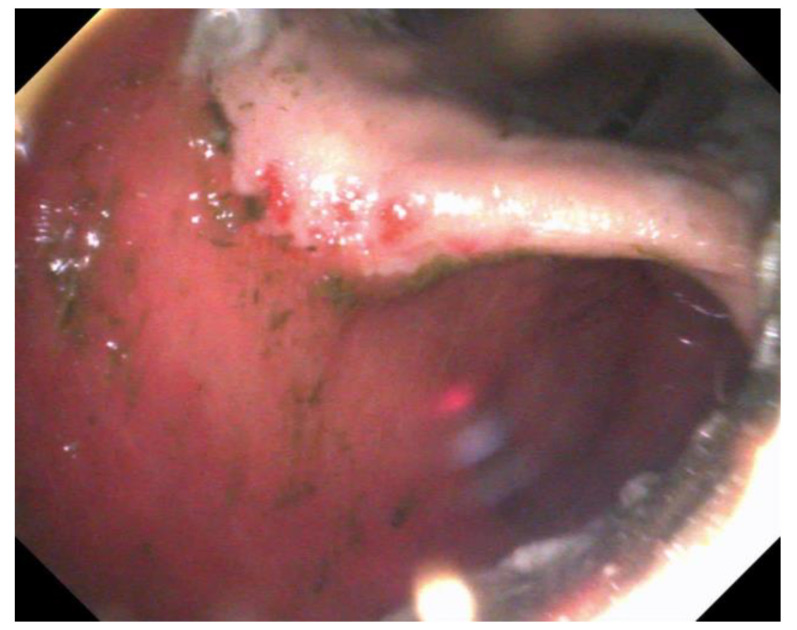
Multifocal, hyperemic lesions of the lesser curvature; grade 2/4 ESGD lesions.

**Figure 3 animals-13-01261-f003:**
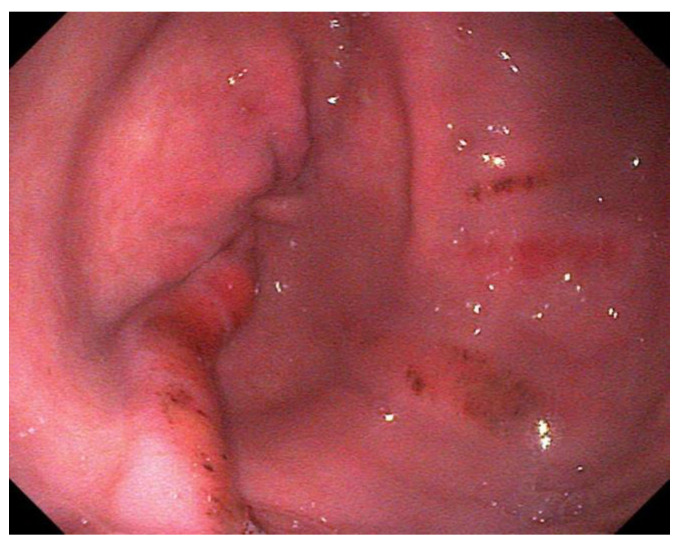
Multiple, linear, flat, hemorrhagic lesions of the pyloric mucosa.

**Figure 4 animals-13-01261-f004:**
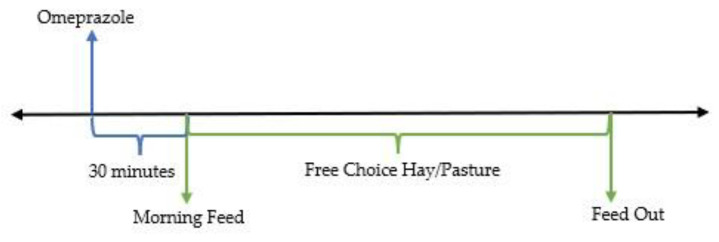
Proposed timing of omeprazole administration and feeding.

**Figure 5 animals-13-01261-f005:**
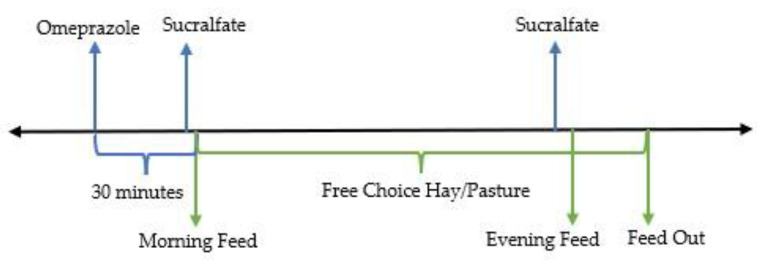
Proposed timing of omeprazole and sucralfate administration and feeding.

## Data Availability

Not applicable.

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
