# Peer review of "Equine Gastric Ulcer Syndrome: An Update on Current Knowledge"

_animals, 2023, doi:10.3390/ani13071261_

Round 1

Reviewer 1 Report

over all this is well-written and comprehensive review. I would suggest the authors include the work of others under the 5.3 NSAID heading showing the association of significant ulceration in both regions of the stomach associated with and NSAID no feed-refeed model. These models are incredibly important for improving our understanding of ulcer diseases in hospitalized horses

Author Response

Thank you for your feedback, we have added a statement to this effect.

Reviewer 2 Report

The Review Article entitle “Equine Gastric Ulcer Syndrome: An update on current knowledge” provides an excellent update of recent literature published after the 2015 the European College of Equine Internal Medicine (ECEIM) consensus statement article published in J Vet Intern Med.

The article is comprehensive, clear and summarised a large body of literature with recommendations and interpretations by the authors clearly indicating their opinions. An Ithenticate® report showed only 5% overlap with current published literature, which is most acceptable.

Some minor suggestions:

Line 108 As it was not “statistically significant:, then no  “association” determined.  Consider amending to

Although not statistically significant, an increased number of caretakers and riders showed a tendency towards an increased risk of EGGD in a mixed Finnish population  [34].

Similar for “A single study comparing treatment timing found an effect of feeding, however this was not statistically significant, likely due to in-440 adequate population size [137].” On line 439

Line 335- One study directly comparing gross  EGGD to histopathological disease showed poor correlation, with 71% of grossly normal  stomachs having mild gastritis, some EGGD classified as mild having severe gastritis, and all severity of EGGD demonstrating mild and moderate gastritis [90

Change “all severity of EGGD to “all severe EGGD cases”

Line 400

In contrast to ESGD, oral omeprazole monotherapy has lower rates of EGGD healing, including 14-25%

 Suggest to replace “including” with “at”

Line 463

treatment might be sufficient of EGGD if adequate…. Consider changing “of” to “for”

Consider putting 8.4 Misoprostal after the acid suppressors

Line 534 A study looking into the concurrent administration of phenylbutazone and omeprazole found a significant increased risk of GI complications in the concurrent treatment group [176].

As this study caused the death of 2 horses and significant GI disease in the other 4 horses in the combination group, please provide more details such as “death of 2/6 from entero or typhlocolitis and colic in 4/6”

The following was taken from: Non-Steroidal Anti-Inflammatory Drugs and Associated Toxicities in Horses

by Flood et al  Animals 2022, 12(21), 2939; https://doi.org/10.3390/ani12212939  and could be referenced in addition to the original study

“There was increased incidence of intestinal adverse events in both phenylbutazone and phenylbutazone/omeprazole groups, with 25% and 75%, respectively, of horses in each group suffering from complications [76]. Intestinal adverse reactions included colic, impactions, diarrhoea, enterocolitis and typhlocolitis resulting in the euthanasia of 2 horses. It is unknown whether these complications were a result of alterations in intestinal motility, exacerbation of NSAID-induced dysbiosis or intestinal inflammation and/or ulceration [76]. Although the dosage of phenylbutazone was double the recommended dosage, these findings are particularly concerning and highlights the importance for further research into concurrent administration of NSAIDs and PPIs in horses.”

Author Response

Thank you for your feedback, we have edited the relevant sections and taken on-board your suggestions. There is one area where we have deviated from your suggestion as follows:

  • Where the suggestion was made to edit line 335 -we have reworded the initial statement to be clearer. The study found that all ‘grades’ of EGGD demonstrated mild and moderate gastritis, not that all the severe EGGD lesions demonstrated this.

Reviewer 3 Report

Dear Authors,

the paper is well written and exhaustive in describing all the aspect of the disease. Some minor corrections:

Literature, number 208, you used the name as surname. COrrect as follow, please: Stucchi L, Zucca E, Serra A, Stancari G, Ceriotti S, Conturba B, Ferro E, Ferrucci F.

Please add these papers in the prevalence section:

Busechian S, Sgorbini M, Orvieto S, Pisello L, Zappulla F, Briganti A, Nocera I, Conte G, Rueca F (2021). Evaluation of a questionnaire to detect the risk of developing ESGD or EGGD in horses. Preventive Medicine, 188: 105285. Doi:10.1016/j.prevetmed.2021.105285

Sgorbini M, Bonelli F, Papini R, Busechian S, Briganti A, Laus F, Faillace V, Zappulla F, Rizk A, Rueca F (2018). Equine Gastric Ulcer Syndrome in adult donkeys: Investigation on prevalence, anatomical distribution, and severity. Equine Veterinary Education, 30(4):206-210. Doi: 10.1111/eve.12747

Author Response

Thank you for this feedback, we have rectified the error and included your suggested references.